# Cadaveric Study on Comparison of Neck Extension Angles for Endotracheal Intubation in Rabbits Using a Rigid and Flexible Endoscope

**DOI:** 10.3390/ani14091270

**Published:** 2024-04-24

**Authors:** Nicharee Luevitoonvechakij, Nithidol Buranapim, Witaya Suriyasathaporn, Pakkanut Bansiddhi, Kanawee Warrit, Veerasak Punyapornwithaya, Wanna Suriyasathaporn

**Affiliations:** 1Graduate Program in Veterinary Science, Faculty of Veterinary Medicine, Chiang Mai University, Chiang Mai 50100, Thailand; nicharee_l@cmu.ac.th; 2Center of Elephant and Wildlife Health, Chiang Mai University Animal Hospital, Chiang Mai University, Chiang Mai 50100, Thailand; pakkanut.b@cmu.ac.th; 3Faculty of Veterinary Medicine, Chiang Mai University, Chiang Mai 50100, Thailand; armvet64@hotmail.com (N.B.); witaya.s@cmu.ac.th (W.S.); kwarrit@gmail.com (K.W.); veerasak.p@cmu.ac.th (V.P.); 4Research Center of Producing and Development of Products and Innovations for Animal Health and Production, Chiang Mai University, Chiang Mai 50100, Thailand; 5Asian Satellite Campuses Institute, Nagoya University, Nagoya 464-8603, Japan

**Keywords:** rabbit, visualized, endotracheal intubation, neck angle, endoscope

## Abstract

**Simple Summary:**

Rabbit endotracheal intubation is often challenging for anesthetists. Employing an endoscope guided by adjusting the angle of head extension can significantly aid anesthetists in accurately locating and executing rabbit endotracheal intubation, reducing the risk of breathing difficulties during the process. This study explores how specific head extension angles impact endotracheal intubation success rates and time taken. The findings recommend utilizing angles of 110 and 120 degrees for optimal rabbit head extension during endotracheal intubation to enhance success rates and efficiency.

**Abstract:**

Endotracheal intubation in rabbits is always challenging due to the unique anatomical conformation. To improve endotracheal intubation success, this study determined the relationship between head placement angles guided by endoscope-assisted visualization techniques and the endotracheal intubation success rate. Thirty-two rabbit cadavers were used in the study. Six veterinary practitioners who had no experience with rabbit endotracheal intubation were randomly assigned to intubate rabbit cadavers using the guidance of either a rigid endoscope (RE) or flexible endoscope (FE), with the head placement angles with an ascending neck at 90, 100, 110, 120 and 130 degrees. The endotracheal intubation completed in 90 s was determined to be a success. The success rates using RE and FE were 97.2% and 95.9%, respectively. The means and standard error of means (SEM) of endotracheal intubation times guided by RE and FE were 53.7 ± 4.68 and 55.2 ± 4.24 s, respectively. Results from survival time analysis show that the five veterinarians successfully intubated the rabbit within 90 s, regardless of the different types of endoscopes. Angle was the only significant factor that affected the endotracheal intubation success. The head placement angle at 110 and 120 degrees had the highest success rate of endotracheal intubation compared to 90 degrees (*p* ≤ 0.05). In conclusion, for inexperienced veterinarians, the success of endotracheal intubation in rabbits, guided by endoscope-assisted visualized techniques regardless of rigid endoscope or flexible endoscope guidance, is improved when the head extension is 110 and 120 degrees.

## 1. Introduction

General anesthesia in exotic animals is always challenging for inexperienced veterinarians. Rabbits are among the most vulnerable species, with a higher death rate during the perioperative period compared to dogs and cats [1]. The narrow and deep oral cavity combined with this species’ large incisors and tongue leads to difficulty visualizing the larynx and glottis, and delicate laryngeal tissue is disrupted during endotracheal intubation, causing unexpected complications [2]. In order to provide inhalation anesthesia and oxygen to rabbits instead of using an endotracheal tube, a facemask technique was used [3]. Nevertheless, it is inadvisable since it has been shown to create hypoxemia and hypercapnia from the relaxation of oropharyngeal tissue, which can result in upper airway blockage and obstruction [4].

For experienced veterinarians, endotracheal intubation in a rabbit is usually performed using a blind technique, and successful endotracheal intubation relies on experience and accustomed head position [5]. To reduce the life-threatening risks from endotracheal intubation, especially for inexperienced veterinarians, many techniques and instruments have been developed, for example, rabbit supraglottic airway devices and wired reinforced endotracheal tubes. The supraglottic airway device, commercially designed for rabbits as V-gel, is an alternative device widely used in human medicine. It is designed to cover the area from the pharynges to the upper esophagus to ensure gas flow and prevent aspiration [6,7,8]. Following the previous investigation, supraglottic airway devices are strictly size-specific. They might not be suitable for rabbits with a weight lower than 2.5 kg and could potentially result in lingual cyanosis in rabbits due to compression of the lingual artery [9,10]. The wired-reinforced endotracheal tube has also been used, and it is known to decrease airway resistance during general anesthesia [11]. Despite its lack of curvature, which facilitates blind endotracheal intubation, this tube’s straight shape enables easier insertion into the trachea [12]. However, these specific devices for rabbit endotracheal intubation have not been available in most parts of the world. 

Many visualization devices, especially the laryngoscope with a specifically designed blade for rabbits, have been used to achieve more accessible endotracheal intubations and improve the success rate of endotracheal intubation in rabbits [13]. However, in some parts of the world where the rabbit laryngoscope blades are not available, the Miller laryngoscope, initially developed as a pediatric laryngoscope for humans, has been used for rabbit endotracheal intubation. However, the difficulty in selecting the fitting blade sizes that are appropriate for visualizing the larynx and epiglottis resulted in laryngeal trauma due to overly attempted endotracheal intubation [13,14,15]. Various sizes of veterinary endoscopes with various light sources are available in animal hospitals. The advantage of using endoscopes for rabbits is the adjustability of the size of endotracheal tube to two-thirds of the tracheal lumen, which is appropriate for endotracheal intubation and verifies proper placement [16]. However, it is unaffordable in every veterinary practice. Therefore, inexpensive rigid endoscopes and flexible fiberscopes from human medicine and available for online purchase could be applied in rabbit endotracheal intubation. 

The best position of the neck angle before endotracheal intubation was reported in previous studies, including mice [17], marmosets [18], and guinea pigs [19], but the hyperextension angle of the rabbit head during intubation is yet to be reported. Therefore, this study aimed to determine factors related to successful endotracheal intubation in rabbits by inexperienced veterinarians using fresh rabbit cadavers. Factors including veterinarians, neck angle position, types of endoscopes, and size of cadavers were evaluated. 

## 2. Materials and Methods

### 2.1. Animals, Endotracheal Tubes, and Study Design 

This study was ethically approved for using cadavers by the Animal Ethics Committee, Faculty of Veterinary Medicine, Chiang Mai University (Ref. No. S1/2564). Thirty-two New Zealand white rabbits, 16 males and 16 females with body weight between 3–5 kg (3.81 ± 0.14), were humanely euthanized. After the death confirmation, the fresh cadavers were immediately used for endotracheal intubation. Two types of inexpensive, online-purchased endoscopes for humans, a household rigid endoscope (USB Digital Endoscope Model: B-B46, RoHS, China) and a flexible fiberscope (USB Wifi Endoscope Model: F140, RoHS, China), were applied. They had different probe characteristics (Figure 1). The rigid endoscope (RE), resembling a pen, was equipped with a camera and light source and primarily designed for otoscopy. The flexible endoscope (FE) featured a long, maneuverable cable fiberscope suitable for accessing confined spaces. The specifications of the rigid endoscope used in this study were as follows: a 15 × 0.8 cm straight composite material with a 3 cm camera length, a lens diameter of 5.5 mm and a 2 cm focus length with 640 × 480 pixels resolution. The flexible endoscope (FE) featured a product size of 5.5 mm × 3.3 cm and a cord length of 1 m, and was constructed from ABS material. It had a focus length of 2 cm, an adjustable light source, and a resolution of 640 × 480 pixels.

A non-cuff wired reinforced endotracheal tube with a 3–3.5 mm diameter, Surgivet^®^ (Smiths Medical, Minneapolis, MN, USA), was used. The tube diameter selection was determined based on the body weight of the rabbit cadavers. The rabbit cadavers with a body weight of between 3–4 kg and over 4 kg were intubated using the 3.0 mm and 3.5 mm diameter endotracheal tubes, respectively. 

The rabbit cadavers were randomly assigned to two groups using different guidance: RE (*n* = 16) and FE (*n* = 16). Six veterinarians with 3 to 7 years of experience in their careers for small animal and exotic animal medicine from Small Animal Hospital, Faculty of Veterinary Medicine, Chiang Mai University, who had no experience with endotracheal intubation on rabbits, either blind or with tool-assisted intubation at the start of the study, were enrolled to perform endotracheal intubation. 

A simple random sampling was used to select a veterinarian, and veterinarians were substituted in the following order when the selected one was unavailable. One cadaver was assigned to a selected veterinarian for the five endotracheal intubation attempts in the five different neck angles. Rabbit cadavers were positioned in sternal recumbency with upright necks to the assigned angles, including 90, 100, 110, 120 and 130 degrees, before endotracheal intubation. The head extension angle was measured by a digital protractor, in which the rabbit’s atlantooccipital joint acted as a fulcrum of extension using the leg as the horizontal axis and the rabbit’s mandible as the vertical axis (Figure 2). The rabbit’s head was placed at the designated angle in each study before the endotracheal intubation process. The endotracheal intubation process began after the arrangement of the neck angle. The endoscope was introduced into the mouth caudoventrally towards the epiglottis. The endotracheal tube was inserted after locating the epiglottis using the side-by-side endotracheal intubation technique. 

To avoid the biased result from the first difficult attempt, each cadaver was randomly assigned to the first attempt angle, followed by the next angles, respectively, until attempts at all angles were performed. All endotracheal intubations had to be performed within 180 s. Unsuccessful endotracheal intubation within 180 s was recorded. The successful endotracheal intubation time was the time from the opening of the cadaver mouth to the confirmation of endotracheal intubation by direct observation. All cadavers were checked for their rigor mortis after successful endotracheal intubation by investigating their epiglottis repositioning before the new test. Cadavers with rigor mortis, indicated by a frozen position of the epiglottis, were excluded from further investigation.

### 2.2. Statistical Analysis

Data were described in means and standard error of means (SEM) for continuous data and percentages for categorical data. Cadaver sizes were determined based on their weights by quartiles as small (<1st quartile), middle (1st to third quartiles), and large (>third quartile). To determine the factors related to endotracheal intubation success within a short time, the success rates within 90 s were analyzed using time-event or survival analyses, including Kaplan–Meier distribution and Cox’s proportional hazard models. The outcome variables included survival time and censor determination. For the failed cases or successful endotracheal intubation within 90 s, the survival times were the duration of endotracheal intubation, and its censor value equaled 0. In contrast, if the endotracheal intubation duration was higher than 90 s, the survival time was 90 s, and its censor was equal to 1. 

The Kaplan–Meier distribution was used to describe the relationship between independent variables and successful endotracheal intubation for univariable analysis. Their statistical significance was defined at *p* < 0.05 using the log-rank test. The independent variables included endoscope types (rigid and flexible), body sizes (small and large), veterinarians (Vet1 to Vet6), and angles of necks (90, 100, 110, 120, and 130). Cox’s proportional hazard models analyzed the final model for factors related to the times of successful endotracheal intubation using the free-entering method. The factor with *p* < 0.15 was allowed to enter and stay in the model. The analysis aimed to assess the hazard ratio (HR) of the specified values compared with a reference value. The assessed HR analogous to risk ratio or relative risk [20] significance level was defined at *p* < 0.05, and the tendency was *p* < 0.1.

## 3. Results

Data of cadavers described, separated by types of endoscopes used, are shown in Table 1. No difference in weights was found between the two groups. From the expected 160 attempts, from 5 times each for the total of 32 cadavers, 16 were excluded due to rigor mortis, including 1, 2, and 3 times missing for 4, 3, and 2 cadavers, respectively. For Vet1, 6 attempts from angles 100° (*n* = 1), 110° (*n* = 2), 120° (*n* = 1), and 130° (*n* = 1) were excluded. The additional excluded data (*n* = 10) were from Vet4 at 110° (*n* = 1), Vet5 at 90° (*n* = 2), 100° (*n* = 2), and 110° (*n* = 1), and Vet6 at 90° (*n* = 1), 100° (*n* = 1), 110° (*n* = 1), and 120° (*n* = 1). Of 144 endotracheal intubations, 137 (95.1%) and 124 (86.1%) endotracheal intubations were successful within 180 and 90 s, respectively. No difference was found between the success rates with RE and FE. Excluding the unsuccessful endotracheal intubations, the minimum, maximum, median, and means (SD) of overall endotracheal intubation times were 12, 172, 41, and 49.9 (29.6) s, respectively.

Kaplan–Meier curves of times to successful endotracheal intubation within 90 s separated by gender (A), endoscope type (B), neck angle (C), veterinarian (D), and body size (E) are shown in Figure 3. The overall median time, or when 50% of attempts were successful endotracheal intubations, was 37 s, with a 95% C.I. at 41 and 46 s. Based on the results from the Log-rank test, significant differences among strata were observed among the veterinarians who performed the endotracheal intubations (Figure 3D), and the trend was observed for angles (*p* = 0.14) and body sizes (*p* = 0.14), as shown in Figure 3C and 3E, respectively. The median times of Vet1 to Vet6 were 83, 38, 38, 29, 52 and 41 s, respectively. The lowest success rate was 58.3% for Vet1, and the highest success rates were 96.6 and 93.3% for Vet4 and Vet3, respectively. The endotracheal intubations using the 90^o^ neck position had the lowest success rates. The median times of the successful endotracheal intubations for 90, 100, 110, 120, and 130 degrees were 50.5, 41, 40, 38, and 40 s, respectively. The success rates at 90 s among angles were 72.4, 85.7, 96.3, 93.3 and 83.3% for 90, 100, 110, 120, and 130 degrees, respectively. Rabbits with large body sizes had the longest time (median = 42 s) for endotracheal intubation time compared to those with small and medium body sizes. The differences among body sizes narrowed as time increased (Figure 3E).

Results from proportional hazard models with *p*-values of less than 0.15 for univariable analysis and less than 0.1 for the final model are shown in Table 2. Body size showed a significant difference in the univariable analysis, in which the small body size had a significantly higher success rate than the large body size (HR = 1.69). In the final model, the overall *p*-values for veterinarians and neck angles were *p* < 0.05 and *p* = 0.08, respectively, while the *p*-value for body size was higher than 0.2. For the veterinarians and the neck angles, results from both univariable analysis and the final model were in the same direction. Compared to Vet1, all other veterinarians had more successful endotracheal intubation rates within 90 s (*p* < 0.05), with Vet4 having 6.25 times the success rate of Vet1. Compared to 90 degrees, neck angles at 110 and 120 degrees had a more successful endotracheal intubation rate within 90 s (HR = 2.22 and 1.78, respectively). No significant differences were observed in both endoscope type and gender.

## 4. Discussion

The laryngoscope with specific blade for rabbits is beneficial for aiding rabbit endotracheal intubation [21]. However, this specific instrument is expensive, costing approximately USD 800 to USD 2000, and might cost more in some countries where it is typically unavailable. Therefore, inexpensive household rigid and flexible fiberscopes for humans, which cost between $5 and $15 and were available everywhere through online purchase in this study, would be the better choice in many parts of the world. Our study revealed that these affordable human endoscopes in both flexible and rigid types could be used to facilitate endotracheal intubation in rabbits. The inexperienced veterinarian could intubate the trachea with the aid of this instrument in a short time and achieve a high success rate when the rabbit’s head was extended to 110–120 degrees. 

Previous research has focused on enhancing rabbit endotracheal intubation for ease and speed, often emphasizing the visibility of the laryngeal structure to reduce the risk of tissue injury and fatalities [22]. Studies reporting the angle for endotracheal intubation in rabbits have been limited to date. In our study, we measured the angle of head extension in rabbits while they were positioned in sternal recumbency, which is recognized as the primary position for rabbit endotracheal intubation. Veterinary practitioners, particularly those with limited experience, are more accustomed to this position than to dorsal recumbency.

Measuring the head extension angle during endotracheal intubation with sternal recumbency in rabbits had yet to be reported at the time of the study. Studies regarding head extension using an adaptable platform to adjust the position of the animal before the endotracheal intubation procedure have been performed in mice [17], marmosets [18], and guinea pigs [19]. The platform for endotracheal intubation supports the animal’s head and neck after the animal has been positioned in dorsal recumbency over the platform. Two types of platforms were used in the referenced studies. The tiltable and fixed platforms had a tiltable edge at the end of the platform, where the animal’s head was positioned. The tiltable part of the platform was angled to 135 degrees, with the floor referenced as the horizontal axis [17,18]. Meanwhile, another study stated that the fixed part of the platform had been angled to 115 degrees, but there was no record of whether the angle of the tiltable edge at the head area had been adjusted [19]. The study on endotracheal intubation angles also extends to human subjects, where the Alexandrou angle of endotracheal intubation (AAI), involving a hospital bed tilted at an angle of 20–30 degrees from the dorsal recumbency position, has been employed to enhance the visualization of vocal cords during orotracheal intubation [23]. The study considered 90 degrees of atlantooccipital joint extension acceptable for endotracheal intubation in humans [24]. The adaptable platform might not be suitable for our study due to the size of the rabbit, especially for the New Zealand White breed, whose average weight for adult rabbits was about 4.08–5.44 kg [25]. Endotracheal intubation with the rabbit positioned in dorsal recumbency has been reported. The advantage of this position is that no assistance is required during endotracheal intubation, as the rabbit is placed on the table edge with the head extended ventrally [14]. To perform endotracheal intubation using this technique, it is recommended to use an over-endoscope intubation approach, as the glottis is located dorsally. The endoscope should act as the style for the endotracheal tube since it needs to advance dorsocaudally [21]. We employed a side-by-side endotracheal intubation method, initially advancing the endoscope into the oral cavity to visualize the epiglottis, followed by insertion of the endotracheal tube through the epiglottis and subsequent withdrawal of the endoscope. Therefore, rabbits are usually positioned in sternal or dorsal recumbency with the head extended caudally for endotracheal intubation. In this study, the rabbits were positioned in sternal recumbency for easier manipulation of head extension angle and angle measurement. 

Hyperextension of the rabbit’s head is the recommended position for endotracheal intubation [5,26]. The present study revealed that hyperextended heads at 110 and 120 degrees are angles appropriate for either rigid or flexible endoscopic guided rabbit endotracheal intubation. The lower-degree angle, which refers to 90 and 100 degrees, or a higher-degree angle (130 degrees) might deviate from the alignment of the oropharynx with the trachea, potentially making the endotracheal intubation process more difficult. Extension of the rabbit’s head over 130 degrees could also damage the skeleton and spinal cord and should be avoided [27]. Therefore, we would suggest 110 and 120 degrees as the preferable angles for endotracheal intubation on rabbits, as the endotracheal intubation success rates and endotracheal intubation times for each of these angles were superior compared to the other angles. Unique anatomical features of the rabbit’s oral cavity may contribute to difficulties in achieving successful endotracheal intubation. These features include long and cutting incisors, the small dimension of the larynx, and obstacles to the direct visualization of the glottis, including the position to align the oropharynx with the trachea [5,14]. A significantly higher endotracheal intubation success rate was found in the small rabbit sizes than in the medium and large rabbit sizes in this study. However, the results from the statistical analysis displayed that the size of the rabbit was not included in the final model of factors related to the time required for successful endotracheal intubation. Less complicated oropharynx structures or confounding factors could be the reason. Endoscope-assisted intubation helps visualization of the epiglottis to ensure precise intubation. However, the limited length of the endoscope may decrease the ability to view the epiglottis in larger-size cadavers. Increasing the sample size might be a viable solution. 

A low success rate with blind endotracheal intubation techniques, requiring multiple attempts, was demonstrated. However, an increased success rate was observed when aided by equipment [28,29]. Affordable online-purchased tools can be used to aid endotracheal intubation, even for veterinarians with no prior experience. Although both types of endoscope can be utilized for rabbit endotracheal intubation, in this study, we found no significant difference in the success rate between using these two devices. According to the previous study, the same type of rigid endoscope was used for endotracheal intubation in rabbits [30,31]; the rigid endoscope can depress the tongue to observe the glottis during endotracheal intubation. The flexible part of a flexible endoscope can also adjust to various angles. However, the quality and durability of the adapted endoscope gradually decreased by the time of the experiment. The decline was evidenced by issues such as the source of light becoming dim, overheating issues, and concerns regarding sterility. These issues might be significant if the tools should be used in living animals. Additionally, there was a camera focus issue, which led to more time-consuming endotracheal intubation. Therefore, the tools should be appropriately stored, moisture and heat should be avoided, and cord bending should be prevented to prolong the tools’ lifetime. The tools should be checked every time before usage, and proper cleaning and sterilization should be considered.

Rabbits are mandatory nose breathers because the epiglottis sits on the soft palate. In this case, the glottis cannot be visualized directly until the soft palate is pushed backward. For this reason, the first attempt to intubate the trachea is always more difficult than the next one, as the glottis can be visualized. This might cause a biased result, as the first angle tested on a new cadaver was more difficult for tracheal intubation compared to the following assessments on the same cadaver. Our design of the randomization for the first attempt angle could help to minimize the expected biased results. This study experimented on cadavers to minimize risk from extensive head extension, tissue trauma, and anesthesia risk. Additional limitations of this study were the limited number of cadavers and the rapid stiffening of some rabbit cadavers, necessitating the removal of data. Further studies should address this limitation by expanding the sample size and applying the study to live rabbits.

This study found that specific head extension angles significantly facilitate endotracheal intubation. The results of the present study revealed the endotracheal intubation success rate and time required. 

## 5. Conclusions

This study examined the relation between the endotracheal intubation time and angle used for head extension before endotracheal intubation and the relevance of the endoscope type used for endotracheal intubation in rabbits. For veterinarians with no experience, the human endoscopes could be applied in rabbit endotracheal intubation with an appropriate head extension angle. The suggested angles for endotracheal intubation in rabbits should be the 110- and 120-degree angles. The type of endoscope used for endotracheal intubation did not significantly decrease the endotracheal intubation time. 

## Figures and Tables

**Figure 1 animals-14-01270-f001:**
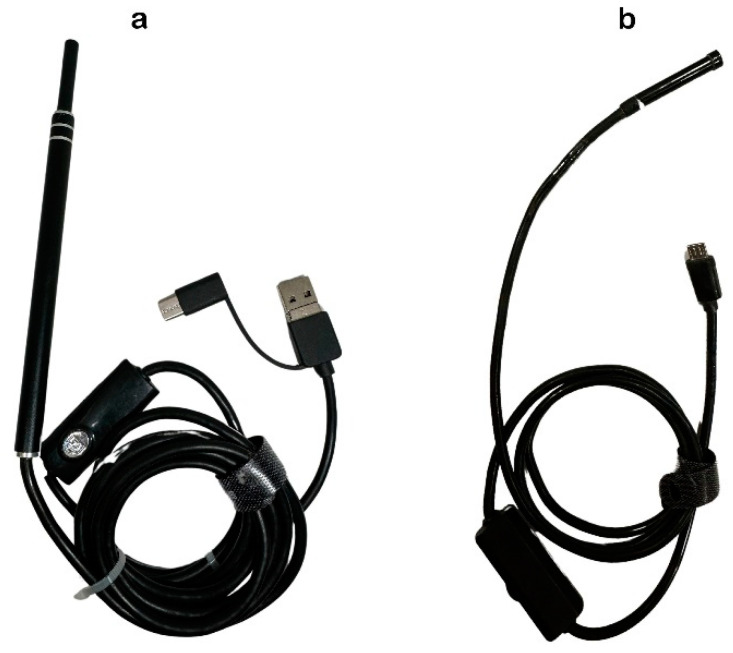
Two types of endoscope. The rigid version (**a**) comprised a 15 × 0.8 cm straight composite material with a 3 cm camera length, a lens diameter of 5.5 mm and a 2 cm focus length with 640 × 480 pixels resolution. The flexible fiberscope (**b**) featured a product size of 5.5 mm × 3.3 cm and a cord length of 1 m, and was constructed from ABS material. It had a focus length of 2 cm, an adjustable light source, and a resolution of 640 × 480 pixels.

**Figure 2 animals-14-01270-f002:**
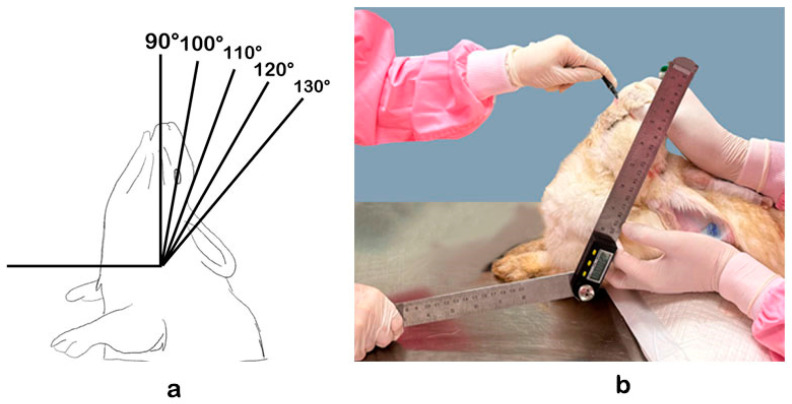
Positioning of the upright neck on dorsal recumbency to the assigned angles, including 90, 100, 110, 120, and 130 degrees (**a**), and the rabbit head extension angle during endotracheal intubation (**b**).

**Figure 3 animals-14-01270-f003:**
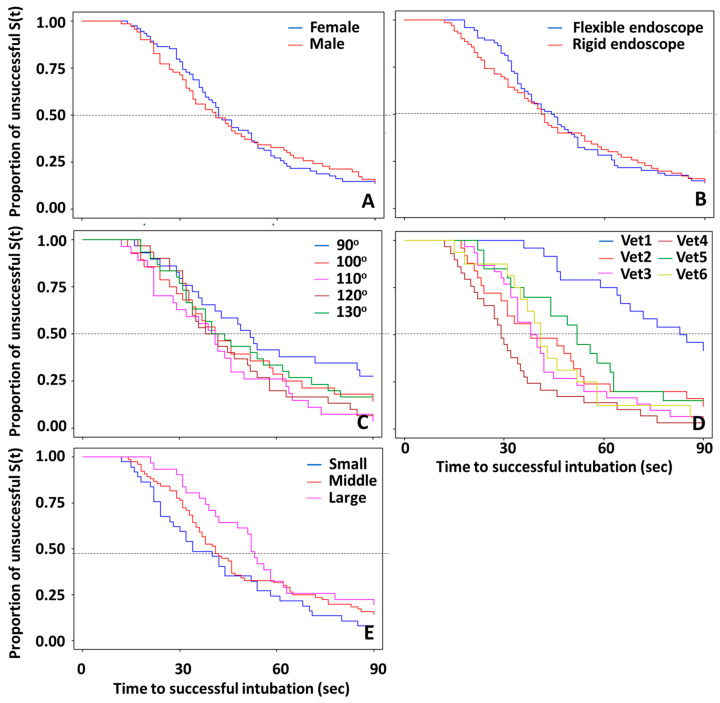
Kaplan–Meier curves of times to successful endotracheal intubation within 90 s separated by gender (**A**), endoscope type (**B**), neck angle (**C**), veterinarian (**D**), and body size (**E**).

**Table 1 animals-14-01270-t001:** Data description of intubation attempts with rigid endoscope (RE) and flexible endoscope (FE). All cadavers (*n* = 32) were intubated five times with different angles, including 90, 100, 110, 120, and 130 degrees. From 160 attempts, 16 endotracheal intubations, 10 for RE and 6 for FE, were excluded due to the rigor mortis of cadavers.

Factor	Rigid Endoscope		Flexible Endoscope	
	*n*	% (mean ± SEM)	*n*	% (mean ± SEM)
Number of cadavers	16	50	16	50
Number of males	5	31.25	10	62.5
Number of females	11	68.75	6	37.5
Body weight (kg)	16	(3.70 ± 0.08)	16	(3.74 ± 0.07)
Endotracheal intubation duration (s)	70	(53.7 ± 4.68)	74	(55.2 ± 4.24)
The success rate at 180 s	70	97.2	74	95.9
The success rate at 90 s	70	85.7	74	86.5

**Table 2 animals-14-01270-t002:** Proportional hazard models, including univariable analyses (Left) and the final model (Right), of factors related to successful endotracheal intubation within 90 s in rabbit cadavers before rigor mortis. The final data include 32 cadavers with 144 endotracheal intubation trials. Variables showed *p*-values of less than 0.15 for univariable analyses and less than 0.1 for the final model.

Factor	Univariable Analysis					Final Model				
	b	SEM	Chi-sq	*p*-Value	HR	b	SEM	Chi-sq	*p*-Value	HR
Veterinarian										
Vet6	1.23	0.38	10.7	<0.01	3.42	1.28	0.38	11.51	<0.01	3.60
Vet5	0.87	0.36	5.7	0.02	2.38	0.83	0.37	5.20	0.02	2.30
Vet4	1.77	0.33	27.8	<0.01	5.80	1.83	0.34	29.54	<0.01	6.25
Vet3	1.31	0.33	15.5	<0.01	3.70	1.28	0.33	14.93	<0.01	3.61
Vet2	1.21	0.34	10.7	<0.01	3.07	1.08	0.34	9.86	<0.01	2.94
Vet1	------------------------------------Reference value ----------------------------------									
Neck Angle										
130°	0.32	0.30	1.15	0.29	1.37	0.27	0.30	0.82	0.36	1.31
120°	0.56	0.29	3.75	0.05	1.76	0.58	0.29	3.86	0.05	1.78
110°	0.71	0.30	5.74	0.02	2.03	0.80	0.30	7.05	<0.01	2.22
100°	0.40	0.30	1.77	0.18	1.49	0.50	0.30	2.71	0.10	1.64
90°	------------------------------------Reference value ----------------------------------									
Cadaver size										
Small	0.52	0.26	3.9	0.05	1.69					
Middle	0.27	0.24	1.27	0.26	1.35					
Large	----------------Reference value -----------------									

## Data Availability

The data presented in this study are available on request from the corresponding author.

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
