# Peer review of "Cadaveric Study on Comparison of Neck Extension Angles for Endotracheal Intubation in Rabbits Using a Rigid and Flexible Endoscope"

_animals, 2024, doi:10.3390/ani14091270_

Round 1
Reviewer 1 Report (Previous Reviewer 2)
Comments and Suggestions for Authors
Dear authors,
thank you for your submission.
Please find below my comments:
line 29: were the vets experienced in using flexible endoscopes? please mention in the manuscript.
line 30: was it a fiberscope or an endoscope? please mention it in the manuscript
Line 47-48: Brodbelt et al. 2008 calculated the throughout perioperative mortality rate and not only during anaesthesia. I believe you should change "during anesthesia" to "during the perioperative period"
Line 73: "otoscopes with laryngoscope blades". what does this mean? is it a laryngoscope blade you place on an otoscope? Please reformulate.
Line 74: "have been used" - please provide the reference
Line 75-80: please split this long phrase into shorter sentences to ease reading
Line 82-83: I believe you refer to the endoscope size (external diameter) and not only to the light source size. Correct me if I am wrong.
Do you refer in the text to endoscopes (camera at the tip of the scope) or to fibroscope (camera at the base followed by a fiber optic to the tip of the scope)? Please specify.
Line 85: otoscopes from human medicine
Line 88-89: please reformulate. the angle is of the head and neck and not of the endo-tracheal intubation
Line 101: again, I have doubts this were not endoscopes but fiberscopes (https://en.wikipedia.org/wiki/Fiberscope)
Line 111-112: Please be more specific
Line 114: Please mention if the vets had experience with endoscopes.
Line 138: please provide the length of the endoscopes. A short endoscope might explain why larger rabbits took longer time to intubate their trachea. Discuss the length also in the discussion section.
Line 150: use the word "introduced into the mouth"
Line 243: "Large" what. It is surprising to see that based on these findings, larger rabbits are more difficult to be intubated. I believe the main reason is the limited length of the endoscope. you should provide the length of the endoscope in M&M
Line 289-290: this definitely needs to be discussed why
Line 307: is it a laryngoscope or an otoscope? please try to be less confusing and be more specific.
line 314-315: ...could "intubate the trachea"...
Line 331: "were used"
Line 344: "IN dorsal recumbency"
Line 357: "Hyperextension of the rabbit's head"
line 358: "the present study reveals"
Line 359: "120 degrees are..."
Line 363: rabbit's head
line 367: "...were SUPERIOR compared to..."
line 367-370: please rephrase to express your thoughts more clearly. try to split the phrase in shorter sentences.
Line 371: "...in THIS study."
Line 379-382: Please rephrase
Comments on the Quality of English Language
Many sentences must be rephrased and become more comprehensible
Author Response
Response to Reviewer 1 Comments
Thank you for giving us the opportunity to submit a revised manuscript. We appreciate the time and effort that you have dedicated to providing your valuable feedback on our manuscript. We are grateful to the reviewers for their insightful comments on the manuscript.
We have been able to incorporate changes to reflect the suggestions provided by the reviewers. We have marked up the changes in yellow highlighted font within the manuscript. Here is a point-by-point response to the reviewers’ comments and concerns.
Comment 1: line 29: were the vets experienced in using flexible endoscopes? please mention in the manuscript.
Answer: We have included the veterinarian experience of using an endoscope from Line 116-120. “Six veterinarians with 3 to 7 years of experience in their careers for small animal and exotic animal medicine from Small Animal Hospital, Faculty of Veterinary Medicine, Chiang Mai University, who had no experience with endotracheal intubation on rabbits, either blind or tool-assisted intubation. at the start of the study, were subjected to perform endotracheal intubation.”
Comment 2 : line 30: was it a fiberscope or an endoscope? please mention it in the manuscript
Answer:Two types of rigid and flexible endoscopes were used in the study. We mentioned them as RE and FE from Line 99-105. We added the word “fiberscope” (Line105) to explain more about the flexible endoscope as your suggestion.
“Two types of inexpensive, online-purchased household …..a long, maneuverable cable “fiberscope” suitable for accessing confined spaces “
Definition of Fiberscope and Endoscope
Fiberscope : a flexible instrument utilizing fiberoptics and used for examination of inaccessible areas
Endoscope: an illuminated usually fiberoptic flexible or rigid tubular instrument for visualizing the interior of a hollow organ or part
https://www.merriam-webster.com › dictionary ›
Comment 3 : Line 47-48: Brodbelt et al. 2008 calculated the throughout perioperative mortality rate and not only during anaesthesia. I believe you should change "during anesthesia" to "during the perioperative period"
Answer : We have changed the sentence to “Rabbits are among the most vulnerable species, with a higher death rate during the perioperative period compared to dogs and cats” as suggested in Line 47-48.
Comment 4 : Line 73: "otoscopes with laryngoscope blades". what does this mean? is it a laryngoscope blade you place on an otoscope? Please reformulate.
Answer : We apologize for the error in our manuscript. We have changed the word otoscope to laryngoscope with a specially designed blade for rabbits for better understanding in Line 73-74.
Comment 5 : Line 74: "have been used" - please provide the reference
Answer : We have included reference 13 : Kim, Yujin, Hee Yeon Jeon, Insook Yang, Ji Hyun Kim, Jae Hee Chung, Ji-Hyeon Ju, Gyeonghun Kim, Jun Won Park, Seung Yeon Oh, Je Kyung Seong, Seung Hyun Oh, and Young-Shin Joo. "Endotracheal Intubation in Rabbits Using a Video Laryngoscope with a Modified Blade." Laboratory Animal Research 38, no. 1 (2022): 24.
Comments 6 : Line 75-80: please split this long phrase into shorter sentences to ease reading
Answer : We have rephrased the sentence in Line 75-80. “However, in some parts of the world where the rabbit laryngoscope blades are not available. Miller laryngoscope, initially developed as a pediatric laryngoscope for humans, has been used for rabbit endotracheal intubation. However, the difficulty in selecting the fitting blade sizes that are appropriate for visualizing the larynx and epiglottis resulted in laryngeal trauma due to overly attempted endotracheal intubation [13-15]”
Comment 7 : Line 82-83: I believe you refer to the endoscope size (external diameter) and not only to the light source size. Correct me if I am wrong.
Answer : We have edited the sentence for better understanding in Line 81-84 “The advantage of using endoscopes for rabbits is the adjustability of the size of endotracheal tube to two-thirds of the tracheal lumen, which is appropriate for endotracheal intubation and verifies proper placement [16] “
Comment 8 : Line 85: otoscopes from human medicine
Answer : We have edited the sentence following the comment on the manuscript in Line 85-86 “fiberscopes from human medicine…”
Comment 9 : Line 88-89: please reformulate. the angle is of the head and neck and not of the endo-tracheal intubation
Answer : We have rephrased the sentence for better understanding in Line 87-89 “ The best position of the neck angle before endotracheal intubation was reported in previous studies, including mice [17], marmosets [18], and guinea pigs [19], but the hyperextension angle of the rabbit head during intubation is yet to be reported .”
Comment 10 : Line 101: again, I have doubts this were not endoscopes but fiberscopes (https://en.wikipedia.org/wiki/Fiberscope)
Answer : Please see Comment 2.
Comment 11 : Line 111-112: Please be more specific
Answer : We have added the specification on selecting the endotracheal size in Line 112-114. “The tube diameter selection was determined based on the body weight of the rabbit cadavers. The rabbit cadavers with a body weight between 3-4 kg, and over 4 kg were intubated using 3.0 mm, and 3.5 mm diameter endotracheal tubes, respectively.”
Comment 12 : Line 114: Please mention if the vets had experience with endoscopes.
Answer : We have already added the sentence “Six veterinarians with 3 to 7 years of experience in their careers for small animal and exotic animal medicine from Small Animal Hospital, Faculty of Veterinary Medicine, Chiang Mai University, who had no experience with endotracheal intubation on rabbits, either blind or tool-assisted intubation at the start of the study, were subjected to perform endotracheal intubation.” in Line 116-120.
Comment 13 : Line 138: please provide the length of the endoscopes. A short endoscope might explain why larger rabbits took longer time to intubate their trachea. Discuss the length also in the discussion section.
Answer : The specification and length of the endoscope have been added. The length of the rigid endoscope is 3 cm camera length has been added in Line 106-107 and the length of the flexible endoscope was in Line 108 which is 3.3 cm. The specification has been added to Figure 1 for more understanding of the figure.
The discussion about the length of the endoscope has been added in Line 381-383.
Comment 14 : Line 150: use the word "introduced into the mouth"
Answer : We have edited the sentence following the comment on the manuscript in Line 156 “The endoscope was introduced into the mouth”
Comment 15 : Line 243: "Large" what. It is surprising to see that based on these findings, larger rabbits are more difficult to be intubated. I believe the main reason is the limited length of the endoscope. you should provide the length of the endoscope in M&M
Answer : Following the comment 13, we have already provided the length of the endoscope in Line 106-108 and the discussion in Line 381-383.
Comment 16 : Line 289-290: this definitely needs to be discussed why
Answer : The discussion about the rabbit body size has been added in Line 381-383 “Endoscope-assisted intubation helps visualization of the epiglottis to ensure precise intubation. However, the limited length of the endoscope may decrease the ability to view the epiglottis in larger-size cadavers.”
Comment 17 : Line 307: is it a laryngoscope or an otoscope? please try to be less confusing and be more specific.
Answer : : We apologize for the error in our manuscript. We have changed the word otoscope to laryngoscope with laryngoscope with specific blade for rabbits for better understanding in Line 313.
Comment 18 : line 314-315: ...could "intubate the trachea"..
Answer : We have edited the sentence following the comment on the manuscript in Line 320-321 “…could intubate the trachea…”
Comment 19 : Line 331: "were used"
Answer : We have edited the sentence following the comment on the manuscript in Line 337 “… platforms were used in the referenced studies.”
Comment 20 : Line 344: "IN dorsal recumbency"
Answer : We have edited the sentence following the comment on the manuscript in Line 350 “…rabbit positioned in dorsal recumbency…”
Comment 21 : Line 357: "Hyperextension of the rabbit's head"
Answer : We have edited the sentence following the comment on the manuscript in Line 363 “Hyperextension of rabbit’s head is the recommended position…”
Comment 22 : line 358 "the present study reveals"
Answer : We have edited the sentence following the comment on the manuscript in Line 364 “…The present study reveals..."
Comment 23 : Line 359: "120 degrees are..."
Answer : We have edited the sentence following the comment on the manuscript in Line 364-365 “…120 degrees are.”
Comment 24 : Line 363: rabbit's head
Answer : We have edited the sentence following the comment on the manuscript in Line 368-369 “Extension of the rabbit’s head…”
Comment 25 : line 367: "...were SUPERIOR compared to..."
Answer : We have edited the sentence following the comment on the manuscript in Line 372 “…angle were superior compared to…”
Comment 26 : line 367-370: please rephrase to express your thoughts more clearly. try to split the phrase in shorter sentences.
Answer : We have rephrased the sentence for better understanding in Line 373-376 “Unique anatomical features of the rabbit's oral cavity may contribute to difficulties in achieving successful endotracheal intubation. These features include long and cutting incisors, the small dimension of the larynx, and hindering the direct visualization of the glottis, including the position to align the oropharynx with the trachea [5, 14].”
Comment 25 : Line 371: "...in THIS study."
Answer : We have edited the sentence following the comment on the manuscript in Line 378 “…large rabbit sizes in this study.”
Comment 27 : Line 379-382: Please rephrase
Answer : We have rephrased the sentence for better understanding in Line 388-390 “Although both types of endoscope can be utilized for rabbit endotracheal intubation. In this study, we found no significant difference in the success rate between using these two devices”
Reviewer 2 Report (New Reviewer)
Comments and Suggestions for Authors
The manuscript by Luevitoonvechakij et al. focuses on the assessment of various neck extension angles and two different laryngoscopes for endotracheal intubation in rabbit cadavers. The study is well designed, all important variables, such as experimentator bias, randomization, animal sex and weight have been considered. The limitations of using cadavers versus living animals have been described, and the economical aspects of using a rigid otoscope compared to a flexible otoscope have been discussed. As a veterinarian, I found the study to be informative for veterinarians working in both small animal medicine and laboratory animal medicine.
I have a few minor comments:
1. In the Methods section (lines 96-98), the weight and sex of the cadavers should be mentioned. The size/weight of the animals is mentioned as a variable further down the line, as well as in the figures, but an absolute number (animals weighing ABC+/-XYZ kg) is missing.
2. Line 104-108: the make and model of the otoscopes would be helpful for the reader who wants to replicate the study or is looking to order an otoscope to use for routine rabbit anesthesias.
3. Table 1: Only males are mentioned, data about females is missing in this aspect?
Author Response
Thank you for giving us the opportunity to submit a revised manuscript. We appreciate the time and effort that you have dedicated to providing your valuable feedback on our manuscript. We are grateful to the reviewers for their insightful comments on the manuscript.
We have been able to incorporate changes to reflect the suggestions provided by the reviewers. We have marked up the changes in red color font within the manuscript. Here is a point-by-point response to the reviewers’ comments and concerns.
Comment 1 : 1. In the Methods section (lines 96-98), the weight and sex of the cadavers should be mentioned. The size/weight of the animals is mentioned as a variable further down the line, as well as in the figures, but an absolute number (animals weighing ABC+/-XYZ kg) is missing.
Answer : The weight and sex of the cadaver have been added as suggested in Line 96-98. The overall body weight has also been added to Line 97-98 “…with the body weight between 3-5 kg (3.81 ± 0.14),…”
Comment 2 : Line 104-108: the make and model of the otoscopes would be helpful for the reader who wants to replicate the study or is looking to order an otoscope to use for routine rabbit anesthesias.
Answer : We have added the model of both types of endoscope in Line 100-101 “rigid endoscope (USB Digital Endoscope Model: B-B046, RoHS, China) and flexible fiberscopes (USB Wifi Endoscope Model: F140, RoHS, China)”
Comment 3 : Table 1: Only males are mentioned, data about females is missing in this aspect?
Answer : The number of female specimens has been added to Table 1.
This manuscript is a resubmission of an earlier submission. The following is a list of the peer review reports and author responses from that submission.
Round 1
Reviewer 1 Report
Comments and Suggestions for Authors
The study aimed to evaluated factors (neck angle position, types of endoscopes, and size of cadavers, veterinarians) related to successful intubation in cadavers of new zealand rabbits by inexperienced veterinarians. In the clinical practice and in the experimental condition the rabbit intubation is the most difficult phase during general anesthesia. Many technique and device are reported in the literature but the mortality rate in rabbit during general anesthesia related to respiratory problem is still high.
The idea of the authors could be interesting but some “ forgetfulness” make the study incomplete and insufficient for publication .
Some consideration:
Title: specify in the title that is a cadaveric study
Introduction
You must provide more references concerning the devices available for rabbits’ intubation
M&M
- The authors luck a careful description of the intubation procedure; for the reader is difficult to understand how the tube and endoscope were introduced. Which technique did they use?
- The angles’ evaluation: using the leg to measure the angle could cause a bias base on the extension of the frontlimb (elbow and shoulder). An objective method to measure the angle should be the use of a support as like the one described in the aforementioned paper 14: Thomas, A A, M C Leach, and P A Flecknell. "An Alternative Method of Endotracheal Intubation of Common Marmosets (Callithrix Jacchus)." Laboratory Animals 46, no. 1 (2012): 71-76.
Results
- Line 199-203: base on the description in this section I don’t understand how many attempts each vet did perform; the authors wrote “Sixty attempted intubations from 6 veterinary participants using RE or FE with randomly assigned angles were performed” therefore were 360 attempts performed? However, the authors in the next sentence and also in the table 1 wrote 144 intubations; what does it mean? It was not clear.
- The “angle results” depends on the results of the single vet; for example which angle did the VET 1 (the one requiring more timing for the intubation) attempt? There are bias in the results.
- A new table is mandatory to describe the distribution of the different angle position for all the animals, clarify which ones were excluded because of the rigor mortis.
Discussion
- Lines 317-321: Angles obtained without platform: the references reported in this section are not appropriate because in these studies the authors used a platform to change the angle of visualization while in the present study the authors changed the angle of the rabbits’ neck.
- 339-344 these comments do not take into consideration the bias associated with the different experiences of each vet.
Comments on the Quality of English Language
Moderate editing of English language required
Reviewer 2 Report
Comments and Suggestions for Authors
Dear authors,
please find below my comments:
Line 16-17: All tracheal intubation techniques use endotracheal methods. Please rephrase or exclude this sentence.
Line 18: throughout the manuscript - please refer to as tracheal intubation or endotracheal intubation, and not only intubation;
Line 23: endotracheal ; no need for a capital letter
Line 23: All mammals have the same anatomical complexity. They all have the same number of laryngeal cartilages, a pharynx.... I believe is not the complexity but the anatomical conformation that differs between mammal species.
Line 34: Angle; use capital letter
Line 35: 110 and 120 degrees...I suppose
Line 48: Also specifically designed V-gel for rabbits do exist. Please rephrase.
Line 56-57: I believe the evidence for not using cuffed ETT is poor. I personally use mostly cuffed ETT and measure the inflation pressure using a manometer to avoid tracheal trauma.
Line 61: Poor evidence of lingual cyanosis
Line 62: "." and not a "," at the end of the line
Line 65-68: Please start a new paragraph if you would like to speak more about the visualization devices. There are also laryngoscope blade specifically designed for rabbits and human use otoscopes described to be used in rabbits.
Line 71: Is the size of the light source or the size of the optic fiber you describe?
Line 82: There is an important aspect you failed to consider in the study designed and you should mention it in the limitations section. Rabbits are mandatory nose breather because the epiglottis sits on the soft palate. In this case the glottis can not be visualized directly until the soft palate is pushed backward. For this reason, the first try to intubate the trachea is always more difficult compared to the next ones as the glottis can be visualized. This will bring a big fault in your results as the first angle tested on a new cadaver will result in a more difficult tracheal intubation compared to the following assessments on the same cadaver . How will you address my concerns?
Line 85-87: I believe you do not need to mention the reason why the rabbits were used before.
Line 87: humanely euthanised
Line 138: please rephrase "unless the intubations end and are recorded as unsuccessful."
Line 199: please rephrase "No difference in weights was found between types of endoscopes."
Line 310: I believe you should give more details regarding the price ratio between the veterinary endoscope and this "affordable" otoscope. Otherwise, the word affordable is subjective.
Line 316-317: Are there old studies but not recent ones?
Line 333-335: Do you have objective reasons for it, or is just an expert point of view from the BSAVA manual? Many practitioners do intubate rabbits in supine position as this allows good extension of the head without the need of a second person to hold the head.
Line 335: Dorsal recumbent and supine position means the same thing.
Line 342-344: Please rephrase. You might say make it more difficult to intubate (as an example)
Line 348: other angles.
Line 348: Anatomy of long and cutting incisors
Line 359: Please mention the elements you would like to compare. Here you only mention blind technique. eg.: comparing blind technique with the use of endoscope
Line 368: Do you want to say successful tracheal intubation?
Line 368: According to a previous study.
Line 370-375: Please rephrase or even split the phrase in shorter sentences for a better clarity.
I hope you find my indication constructive.
Comments on the Quality of English LanguageSome sentences need to be rephrased for a better clarity.